# Peer review of "Application of C-Terminal Clostridium Perfringens Enterotoxin in Treatment of Brain Metastasis from Breast Cancer"

_cancers, 2022, doi:10.3390/cancers14174309_

Round 1

Reviewer 1 Report

The authors have done a good review of all aspects concerning the application of CPE in treating breast cancer metastasis to the brain, this review enjoyed reading it and would like to make recommendations on some minor revisions:

1. Since this is a review focusing on treating brain metastasis from breast cancer, the authors may consider citing more studies specifically investigating how breast cancer metastasis in the brain can be treated by CPE and emphasize this in the title of section 4, the current manuscript has very few references related to this area.

2. The authors should check if reference 34 is appropriately used to support line 185-187 since this reviewer can not find any related information in Ref 34 whereas Ref 28 seems to be relevant. 

Author Response

  1. The authors have done a good review of all aspects concerning the application of CPE in treating breast cancer metastasis to the brain, this reviewer enjoyed reading it and would like to make recommendations on some minor revisions:

Thank you!

  1. Since this is a review focusing on treating brain metastasis from breast cancer, the authors may consider citing more studies specifically investigating how breast cancer metastasis in the brain can be treated by CPE and emphasize this in the title of section 4, the current manuscript has very few references related to this area.

We agree with the reviewer and have added more information under section 4 citing properly documented studies on treatment options for breast cancer metastasis in the brain and why the integration of c-CPE can enhance drug delivery and target therapy.  (lines 290 -298) .

Reviewer 2 Report

The review submitted by Banga and collaborators presents an interesting rationale for the use of the C-Terminal Clostridium Perfringens Enterotoxin (CPE) in the treatment of brain-metastatic breast tumors. The author's argumentation highlighted the relevance of CPE binding to Claudin 4, which is overexpressed in breast and other tumors, but virtually absent in brain and/or non-tumoral cells.  This association might induce cell death by disrupting selective cellular permeability and might also serve as a drug-delivered system by conjugating CPE with drug-associated nanoparticles.  This is a highly relevant approach for future therapeutic strategies.

 Nevertheless, I consider that the following minor points should be addressed/clarified:

 1- Figure 2: It would be clearer if the processes represented by the numbers 1-7 were explained in the legend. Without a better explanation of the ideas, it is hard to understand, for example, the presence of DC and CD8+ cells in the cytoplasm of a tumor cell.

2- Figure 3: Legend and or text must be improved addressing the following questions: Is the heatmap presented representative of new/unpublished data? How was it constructed? Which is the source of the information provided?

3-      References supporting the following affirmations must be added to the text:

-          Page 2, lines 78-80: “Tubular arrangement of tumor epithelial cells is associated with Claudins as Claudin deletion mutants lead to 79 tumors with decreased tubular arrangement.”

-          Page 4, lines 140-141: “Tumorigenesis, progression, and pathology of breast cancer have been linked to different expressions of Claudins and Claudin-4 has proved very powerful in predicting survival.

Author Response

Dear Reviewer,

Thank you very much for your careful reading and for providing valuable comments to improve the quality of the manuscript.

We really appreciate your valuable comments.

Hereby we are providing the response to the comments.

 1- Figure 2: It would be clearer if the processes represented by the numbers 1-7 were explained in the legend. Without a better explanation of the ideas, it is hard to understand, for example, the presence of DC and CD8+ cells in the cytoplasm of a tumor cell.

  • Thank you for this suggestion. This information has been added into the legend. Also, the dendritic cells were intended to be in the extracellular space.. We apologize for the error and have corrected this. (line 90)

2. Figure 3 legend  Can more details about what has been performed here, and what is being examined please be included.  E.g. Is this protein, or gene expression? How was this data obtained?

  • Thank you for this comment. The data was obtained from a review of several papers that discuss the claudin expressions in cancers.  The references are now present in the manuscript and included in the legend, which has also been edited to improve clarity (lines 164-167). An extra paragraph delving into more detail has also been added to improve clarity (lines 141 -156

3-      References supporting the following affirmations must be added to the text:

-          Page 2, lines 78-80: “Tubular arrangement of tumor epithelial cells is associated with Claudins as Claudin deletion mutants lead to 79 tumors with decreased tubular arrangement.”

  • Thank you for this observation. This information has been given the appropriate reference (line 81).

-          Page 4, lines 140-141: “Tumorigenesis, progression, and pathology of breast cancer have been linked to different expressions of Claudins and Claudin-4 has proved very powerful in predicting survival. 

  • Thank you for this observation. This information has been given the appropriate reference (lines 183-185).

Reviewer 3 Report

This manuscript would benefit from a schematic representation of the various domains of CPE, as functional and structural domains are referred to at various points in the text, and this schematic would help with visualising this.

Figure 1 legend – please define what ZO-1 is.

Figure 1 – this figure would benefit from including some of the information included in the text (e.g. labelling of the ECL domains).

3.1 Brain Metastasis Treatment Options

* For the cases (98,000 to 170,000 new diagnoses) – are these number annual numbers? This will need to be clarified in text.

Line 146  Should be Figure 3.

Figure 3 legend  Can more details about what has been performed here, and what is being examined please be included.  E.g. Is this protein, or gene expression? How was this data obtained?

Line 150  please provide references for CPE treatment being shown to have claudin-4 dependence in breast cancer.

For lines 150 – 153  what cell types was this study performed in?

Please expand upon CPE being able to act as a claudin modulator to impermanently open TJs for improved delivery of drugs (line 153 – 154), and provide a discussion of specific examples.

Some additional detail is required when discussing some of the studies.  For example, which breast cancer model is being used in reference 34, and what control was being examined (lines 185-187)?

Some of the statements require supporting references. For example, claudin-4 expression in normal areas of the CNS is negligible restricting apoptosis exclusively to brain tumor cells and hindering tumor growth (lines 190-192, particularly the statement relating to expression).

An additional study (Ebihara et al. 2006. J Pharmacol Exp Ther 316: 255-60 https://pubmed.ncbi.nlm.nih.gov/16183701/ ) should be discussed in this manuscript.

Author Response

Dear Reviewer,

Thank you very much for going through the manuscript thoroughly  We really appreciate your valuable comments and these suggestions really improve the quality of our paper

This manuscript would benefit from a schematic representation of the various domains of CPE, as functional and structural domains are referred to at various points in the text, and this schematic would help with visualising this.

  • Thank you for this suggestion. A new figure (Figure 5) has been added to the manuscript. (Lines 211) .

Figure 1 legend – please define what ZO-1 is.

  • Thank you for this suggestion. ZO-1 refers to zonal occludens -1, a tight junction protein that also modulates cell-cell adhesion. This has been added into the text  (line 74).

Figure 1 – this figure would benefit from including some of the information included in the text (e.g. labelling of the ECL domains).

  • Thank you for this suggestion. A new figure (Figure 1B) has been added to the manuscript detailing this information (lines 69)

3.1 Brain Metastasis Treatment Options

* For the cases (98,000 to 170,000 new diagnoses) – are these number annual numbers? This will need to be clarified in text.

  • These numbers are annual and have been clarified in the text (lines 112)

Line 146 ïƒ  Should be Figure 3.

Thank you for this observation. We have included this correction in the manuscript (lines 146)

Figure 3 legend ïƒ  Can more details about what has been performed here, and what is being examined please be included.  E.g. Is this protein, or gene expression? How was this data obtained?

  • T

    Thank you for this comment. The data was obtained from a review of several papers discussing claudin expressions in cancers.  The references are now present in  the manuscript and included in the legend, which has also been edited to improve clarity(lines 171-174). An extra paragraph delving into more detail has also been added to improve clarity (lines 141 -156)

Line 150 ïƒ  please provide references for CPE treatment being shown to have claudin-4 dependence in breast cancer.

  • Thank you. The reference describing how Claudin-4 expression sensitizes breast cancer cells to CPE treatment has been added (line 183)

For lines 150 – 153 ïƒ  what cell types was this study performed in?

  • Thank you for this correction, we have included cell types and reorganized the paragraph for a better flow (line 176)

Please expand upon CPE being able to act as a claudin modulator to impermanently open TJs for improved delivery of drugs (line 153 – 154), and provide a discussion of specific examples.

  • Thank you for this suggestion. The C-terminal fragment is the non-cytotoxic portion which allows the entry of drugs without damaging the plasma membrane, thus the line has been corrected on 153 and rewritten in the c-CPE portion of the text. More examples showing c-CPE usage in drug delivery is also included (lines 220-260)

Some additional detail is required when discussing some of the studies.  For example, which breast cancer model is being used in reference 34, and what control was being examined (lines 185-187)?

  • Thank you for this correction, we have included the breast cancer model used and controls, this study was performed in. (line 202-205)

Some of the statements require supporting references. For example, claudin-4 expression in normal areas of the CNS is negligible restricting apoptosis exclusively to brain tumor cells and hindering tumor growth (lines 190-192, particularly the statement relating to expression).

  • Thank you for this observation. We have included a references for the study we reviewed to provide more insight to scientific research that supports this statement. (line 266)

An additional study (Ebihara et al. 2006. J Pharmacol Exp Ther 316: 255-60 https://pubmed.ncbi.nlm.nih.gov/16183701/ ) should be discussed in this manuscript.

  • Thank you for this observation. We have reviewed this paper and included information revealing how c-CPE treatment significantly improved cancer therapy by increasing localization and effectiveness, and decreasing toxicity in cancer cells high in Claudin-4. (lines 246-250)